# Accuracy of sarcopenia screening tools among older adults with stroke undergoing inpatient rehabilitation

**Kanokphol Supasirimontri[1], Vilai Kuptniratsaikul[1]*, Metita Wiwitkul[2]**

**1** Department of Rehabilitation Medicine, Faculty of Medicine Siriraj Hospital, Mahidol University, Thailand,
**2** Division of Occupational Therapy, Department of Rehabilitation Medicine, Faculty of Medicine Siriraj Hospital, Mahidol University, Thailand

* vilai.kup@mahidol.ac.th

## Abstract

### Background

Appropriate screening tools for sarcopenia are available for older adults with stroke undergoing inpatient rehabilitation, however how well these tools predict and eventually diagnose sarcopenia remain unknown.

### Objective

To investigate the accuracy of five index tools including calf circumference (CC), the strength, assistance with walking, rising from a chair, climbing stairs and falls (SARC-F), the CC combined with SARC-F (SARC-CalF), the 5-item and 7-item Mini Sarcopenia Risk Assessment (MSRA-5 and MSRA-7) for screening sarcopenia among older adults with stroke undergoing inpatient rehabilitation according to the guideline from the Asian Working Group for Sarcopenia 2019.

### Methods

Stroke patients aged sixty years and older who admitted for inpatient rehabilitation were included between April 2023 and June 2024. In this study, sarcopenia was defined by low skeletal muscle mass and low grip strength. Skeletal muscle mass was evaluated using a bioelectrical impedance analysis. Grip strength was assessed using a digital handheld dynamometer. The five index tests were applied. Diagnostic indices and areas under the receiver operating characteristic curves (AUC) were analyzed. Youden's index was used to determine the optimal cut-off values for the index tests.

### Results

One-hundred and fourteen patients with stroke were included (55.26% male) with mean age of 70.39 years. The prevalence of sarcopenia was 50.88%. The AUC

**Data availability statement:** All data supporting the findings of this study are available upon the link https://zenodo.org/records/19565382 or can be accessed via DOI 10.5281/zenodo.19565382.

**Funding:** This study was supported by the Faculty of Medicine Siriraj Hospital, Mahidol University, Bangkok, Thailand (Grant Number R016631042.). The funders had no role in study design, data collection and analysis, decision to publish, or preparation of the manuscript.

**Competing interests:** The authors have declared that no competing interests exist.

with 95% confidence interval of CC, SARC-F, SARC-CalF, MSRA-5 and MSRA-7 were 0.873 (0.797–0.928), 0.653 (0.558–0.740), 0.832 (0.751–0.896), 0.641 (0.546–0.729), and 0.626 (0.530–0.715), respectively. The optimal cut-off points of the CC were < 33 cm for men, and < 32 cm for women, and the optimal cut-off point of SARC-CalF was ≥ 16.

## Conclusions

The CC and SARC-CalF were demonstrated appropriate diagnostic performance for screening sarcopenia in older adults with stroke undergoing inpatient rehabilitation.

---

## Introduction

Sarcopenia is characterized by a generalized loss of skeletal muscle mass and strength, and is associated with an increase risk of adverse outcomes including physical impairment, falls, fractures, and mortality [1]. Stroke is a common condition among older adults, and contributes substantially to healthcare costs as well as physical and functional impairments [2]. Sarcopenia can occur across all phases of stroke [3]. A previous meta-analysis found that half of patients with stroke develop sarcopenia [4]. The prevalence of sarcopenia increases from pre-stroke period through the acute phase and continues to rise during the first month after stroke [5]. Moreover, the prevalence of sarcopenia tends to be higher in the chronic phase of stroke than in the acute phase [3]. Sarcopenia among patients with stroke can adversely affect functional outcomes such as ambulation, activities of daily living, and swallowing function [6,7].

According to the Asian Working Group for Sarcopenia 2019 (AWGS2019), the diagnostic criteria for sarcopenia in Asian populations include reduced muscle mass accompanied by either low muscle strength or poor physical performance [8]. Muscle mass is assessed using dual-energy x-ray absorptiometry or bioelectrical impedance analysis, whereas muscle strength is commonly measured using handheld dynamometer. However, the high cost and limited accessibility of these tools limit their widespread use in routine clinical assessment of sarcopenia [9].

Currently, several tools are available for screening or diagnosing sarcopenia, including calf circumference (CC), the strength, assistance with walking, rising from a chair, climbing stairs and history of falls (SARC-F), the combined test between the CC and SARC-F (SARC-CalF) [8], and the 5-item and 7-item Mini Sarcopenia Risk Assessment questionnaires (MSRA-5 and MSRA-7) [10]. However, most of these tools have been validated primarily in healthy community-dwelling populations or in patients with specific medical conditions. Among patients with stroke, previous studies have applied these screening tools mainly in outpatient settings [11–12], without specifically focusing older adults [11–12]. To the best of our knowledge, screening tools for sarcopenia are available among older adults with stroke undergoing inpatient rehabilitation, but their diagnostic performance remains uncertain. Therefore,

the objective of this study was to evaluate the diagnostic accuracy of these selected index tests for screening sarcopenia in this specific population.

## Materials and methods

### Study design and participants

We conducted a diagnostic accuracy study between April 2023 and June 2024 at the rehabilitation ward, Faculty of Medicine Siriraj Hospital, Mahidol University, Bangkok, Thailand. The inclusion criteria were:(1) the stroke patients aged 60 years or older admitted for rehabilitation; (2) Stable medical and neurological conditions were defined as the absence of acute events requiring urgent intervention and eligibility for inpatient rehabilitation. In addition, functional and cognitive readiness were evaluated by the ability to maintain an upright position for a minimum of two hours and intact recent memory within the past twenty-four hours [13]; and (3) able to follow at least a one-step command to ensure full cooperate during functional assessments. The exclusion criteria included the following conditions: (1) inability to communicate in the Thai language or the presence of communication impairments, including all types of dysphasia; (2) the presence of an implanted pacemaker or cochlear implantation [14]; (3) significant cognitive impairment, defined as a Thai Mental State Examination (TMSE) score < 24 [15]; (4) limb amputation [16]; (5) active medical conditions including active tuberculosis, autoimmune disease, organ transplantation, acquired immunodeficiency syndrome, end stage renal disease or malignancy [16]; (6) the presence of prostheses or metallic implants in the extremities (e.g., joints, plates) [14]. All participants were evaluated for sarcopenia using both the reference standard and index tests within the first three days of admission. This study was approved by the Institutional Review Board, Faculty of Medicine Siriraj Hospital (COA no. Si 257/2023). The written informed consent was obtained from all participants. The study adhered to the principles outlined in the Declaration of Helsinki.

### Demographic characteristics

Demographic data were collected through face-to-face interview with eligible participants and by reviewing their medical records. Stroke-related data included comorbidities, stroke characteristics, and the current Functional Ambulation Categories (FAC) at admission. The FAC is a six-level scale that classifies walking ability according to the level of assistance required. The scale ranges from 0, indicating inability to walk or the need for assistance from two or more people) to 5 indicating independent walking on all surfaces) [17].

### Reference standard to diagnose sarcopenia

The reference standard for diagnosing sarcopenia was based on the criteria of the AWGS 2019, which are tailored for Asian populations. According to these criteria, sarcopenia is defined by the presence of both low muscle mass—defined as appendicular skeletal muscle mass index < 7.0 kg/m$^2$ for men and < 5.7 kg/m$^2$ for women—and low muscle strength, defined as mean handgrip strength < 28 kg for men and < 18 kg for women [8]. The physical performance tests were not included in the definition of sarcopenia in this study, as older adults with stroke often have mobility or ambulation impairments that limit the feasibility of these assessments. Therefore, sarcopenia on this study was diagnosed based on the combined presence of low muscle mass and low muscle strength [8,18,19].

### Measurement of muscle mass and handgrip strength

Appendicular skeletal muscle mass was assessed using a bioelectrical impedance analysis device (InBody S10; InBody Co. Ltd., Korea). Participants lay in the supine position on a bed with appropriate posture and electrode placement. If any error messages or irregular impedance signals appeared on the device screen, the participants' hands and feet were cleaned. In such cases, the measurement was repeated until valid results were obtained [14] To minimize the effect of

food intake on the measurements, assessments were performed at least three hours after the last meal [19]. Then, appendicular skeletal mass index was computed by appendicular skeletal mass (kg) / height (m)$^2$.

Additionally, handgrip strength was assessed by an experienced occupational therapist using a digital handheld dynamometer (Jamar Plus, Performance Health Supply, Inc., United States) [8]. Participant were seated with elbow flexed at ninety degrees and wrists in a neutral position. Handgrip strength of each participant was assessed bilaterally to account for stroke-related motor impairments including unilateral and bilateral upper limb weakness. Three measurements were obtained for each hand, and the mean value of the three trials was calculated for each side. The higher mean handgrip strength was selected as the representative value for subsequent analyses [8,12,20].

## The index tests

The CC were measured bilaterally using a non-elastic tape. While in the supine position, participants maintained a 90-degree knee flexion with relaxed ankles and feet. The tape was wrapped around the largest part of the calf, perpendicular to the longitudinal axis of the leg. The maximum value from each side was recorded, and the larger of the two was selected for analysis [21]. The original cut-off values for sarcopenia screening were < 34 cm for men and < 33 cm for women [8].

The SARC-F consists of of five questions including the topics of strength, assistance with walking, rising from a chair, climbing stairs and falls. Each item is scored on a 3-point scale (0–2), with total scores ranging from 0 to 10. A total score of 4 or more indicates possible sarcopenia [8]. This study used the validated Thai version of SARC-F, which has demonstrated appropriate psychometric properties [9].

The SARC-CalF is the combined test between the CC and SARC-F. The total score of SARC-F (score 0–10) is added to a CC score of either 0 or 10. For men and women, a CC > 34 cm and > 33 cm is scored as 0, respectively, while ≤ 34 cm and ≤ 33 cm are scored as 10. This scoring system yields a total score ranging from 0 to 20. A score of ≥11 indicates possible sarcopenia [8,22].

The Mini Sarcopenia Risk Assessment includes two versions: a 5-item (MSRA-5) and a 7-item questionnaire (MSRA-7). Both versions have been validated in Thai [9]. The maximum possible scores of the 5-item and 7-item Mini Sarcopenia Risk Assessment were 60, and 40, respectively. The original cut-off values for sarcopenia screening were ≤ 45 for the MSRA-5, and ≤ 30 for the MSRA-7, indicating an increased risk of sarcopenia [10].

## Statistical analysis

Sample size was calculated based on a previously published equation for diagnostic test studies [23]. According to previous study, the prevalence of sarcopenia among patients with stroke was estimated at 50% [4]. Reported sensitivities for the index tests were as follows: the CC in men: 85% [21], and the CC in women: 91% [21]; the SARC-F: 82.9% [24]; the SARC-CalF: 94.7% [25], the MSRA-5: 90.2% [26] and the MSRA-7: 86.9% [26] respectively. Using a Z value of 1.96 (for $\alpha = 0.05$) and an allowable error of 0.10, the sample size calculation was based on the lowest sensitivity (the value of the sensitivity of the SARC-F: 82.9%) [24], which yield the largest sample size requirement to ensure the adequate power for all index tests. Therefore, the final estimated sample size was 114 participants.

Categorical variables were summarized using frequency and percentage, while continuous variables were presented as mean and standard deviation. Group comparisons were performed using the chi-square test, Fisher's exact test, or independent samples t-test, as appropriate. Diagnostic performance indices were calculated for each index test and included:sensitivity, specificity, accuracy, positive predictive value, negative predictive value, positive likelihood ratio, negative likelihood ratio, and area under the receiver operating characteristic curve (AUC) with 95% confidence interval (CI). Based on the values, AUCs were interpreted as indicating high (>0.90), moderate (0.70–0.90), low (0.50–0.70), and no discriminatory ability (0.5, equivalent to random chance) [27]. Moreover, optimal cut-off values for the screening tools were determined using Youden's index calculated as sensitivity + specificity − 1 [28]. All statistical analyses were conducted using

MedCalc for Windows, Version 15.0 (MedCalc Software, Ostend, Belgium), and PASW statistics for Windows, version 18.0 (SPSS Inc., Chicago, IL., USA). A p values below 0.05 were considered as statistically significant.

## Results

According to the Fig 1, the total of 232 participants were screened for eligibility. Of these, 118 were excluded for the following reasons: age less than 60 years (n = 61), cognitive impairment (n = 25), communication difficulties (n = 16), presence of metallic implants in extremities (n = 7), active cancer (n = 4), implanted pacemaker (n = 2), refusal to participate (n = 2), and end stage renal disease (n = 1).

The final study population included 114 participants (63 men and 51 women) with a mean age 70.39 (SD 7.27) years old. The prevalence of sarcopenia is 50.88%. Eighty-seven patients (76.32%) had ischemic stroke. Participant characteristics, stratified by sarcopenia status according to the AWGS2019 criteria, are summarized in Table 1. No statistically significant differences were observed between the sarcopenia and non-sarcopenia groups except for age, body mass index (BMI), and onset of stroke to rehabilitation admission. Participants in the sarcopenia group were generally older, had lower BMI, and more often presented in the subacute phase of stroke. In contrast, a greater proportion of patients in the non-sarcopenia group were in the acute phase of stroke onset. Most participants with the FAC scores of 0–2 (ranging from 87.50% to 96.55%) were unable to ambulate independently at the time of admission.

Based on the reference standard defined by the AWGS2019, the diagnostic performance of all index tests was assessed using the receiver operating characteristic curve analysis among older adults with stroke undergoing inpatient rehabilitation (Fig 2). The values of AUC (95% CI) of the CC and the SARC-CalF were 0.873 (0.797–0.928), and 0.832 (0.751–0.896), respectively. These values indicate moderate discriminatory ability (AUC between 0.7 and 0.9). Nevertheless, the values of AUC of the other screening tools were lower; the SARC-F, MSRA-5 and MSRA-7 were 0.653

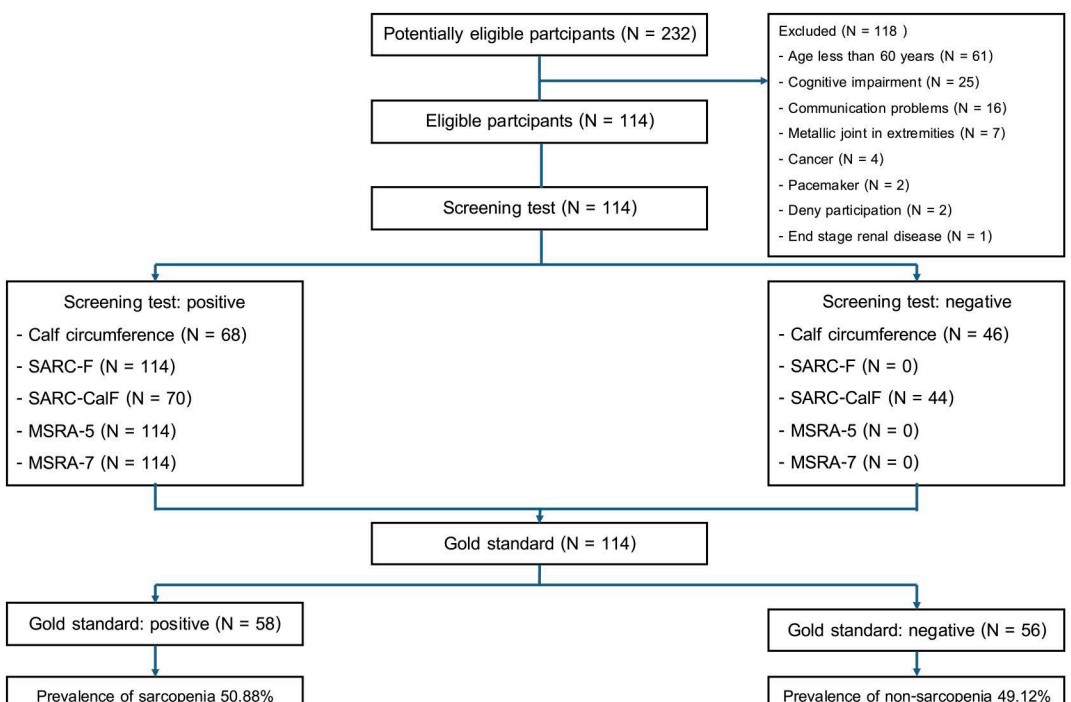

**Fig 1. Flow diagram for screening and diagnosing sarcopenia among older adults with stroke undergoing inpatient rehabilitation.**

**Table 1. Characteristics of participants according to the Asian Working Group for Sarcopenia 2019 definition.**

| Characteristics | Sarcopenia (N = 58) | Non-sarcopenia (N = 56) | p |
|---|---|---|---|
| Age (years)[1] | 72.21 (8.03) | 68.50 (5.90) | 0.006[a] |
| Body mass index (kg/m²)[1] | 21.18 (3.30) | 24.53 (3.29) | <0.001[a] |
| Male[2] | 33 (56.90%) | 30 (53.57%) | 0.721[b] |
| Hypertension[2] | 45 (77.59%) | 47 (83.93%) | 0.391[b] |
| Diabetes mellitus[2] | 28 (48.28%) | 24 (42.86%) | 0.561[b] |
| Dyslipidemia[2] | 35 (60.34%) | 39 (69.64%) | 0.298[b] |
| Atrial fibrillation[2] | 8 (13.79%) | 5 (8.92%) | 0.414[b] |
| Cardiovascular disease[2] | 6 (10.34%) | 7 (12.50%) | 0.717[b] |
| Current smoking[2] | 4 (6.89%) | 4 (7.14%) | 1.000[c] |
| Current alcohol drinking[2] | 1 (1.72%) | 5 (8.92%) | 0.110[c] |
| Type (ischemic stroke)[2] | 43 (74.13%) | 44 (78.57%) | 0.578[b] |
| History (first time stroke)[2] | 42 (72.41%) | 43 (76.79%) | 0.592[b] |
| Onset of stroke to rehabilitation admission[2] <br>• Acute phase (within 30 days) <br>• Subacute phase (within 3 months) <br>• Chronic phase (more than 3 months) | <br>11 (18.96%) <br>28 (48.28%) <br>19 (32.76%) | <br>23 (41.07%) <br>16 (28.57%) <br>17 (30.36%) | 0.023[b] |
| Side of hemispheric lesion (right)[2] | 39 (67.24%) | 35 (62.50%) | 0.336[b] |
| Functional ambulatory category[2] | | | 0.189[b] |
| • Category 0 | 31 (53.45%) | 22 (39.29%) | |
| • Category 1 | 22 (37.93%) | 22 (39.29%) | |
| • Category 2 | 3 (5.17%) | 5 (8.92%) | |
| • Category 3–5 | 2 (3.45%) | 7 (12.50%) | |

1 mean (standard deviation), ²number (%).

a independent samples t-test, ᵇ chi-square test, ᶜ fisher's exact test.

(0.558–0.740), 0.641 (0.546–0.729), and 0.626 (0.530–0.715), respectively. These tools demonstrated low discriminatory ability of diagnostic indices with the value of AUC between 0.5 and 0.7.

Table 2 presents the diagnostic indices of all tests using their original cut-off values. Those were CC sensitivity, 90.91% (75.67%−98.08%) for men, 84.00% (63.92%−95.46%) for women; the SARC-CalF, 89.66% (78.83%−96.11%). While the values of specificity of the CC, 73.33% (54.11%−87.72%) for men; 65.38% (44.33%−82.79%) for women participants, and the SARC-CalF, 67.86% (54.04%−79.71%). The SARC-F, MSRA-5, and MSRA-7 each demonstrated 100% (93.84%−100%) sensitivity but 0% (0%−6.38%) specificity, meaning they identified all true positives but failed to detect true negatives. As a result, negative predictive value and negative likelihood ratio could not be calculated due to the absence of true negatives.

Additionally, Table 3 provides revised optimal cut-off points based on Youden's index, which showed: CC < 33 cm for men, and < 32 cm for women, while the optimal cut-off points of ≥ 16scores in the SARC-CalF yielded the highest diagnostic performance.

## Discussion

Our study demonstrated that the CC and the SARC-CalF exhibited acceptable diagnostic accuracy for detecting sarcopenia among older stroke inpatients undergoing rehabilitation. In contrast, the other tools including the SARC-F,

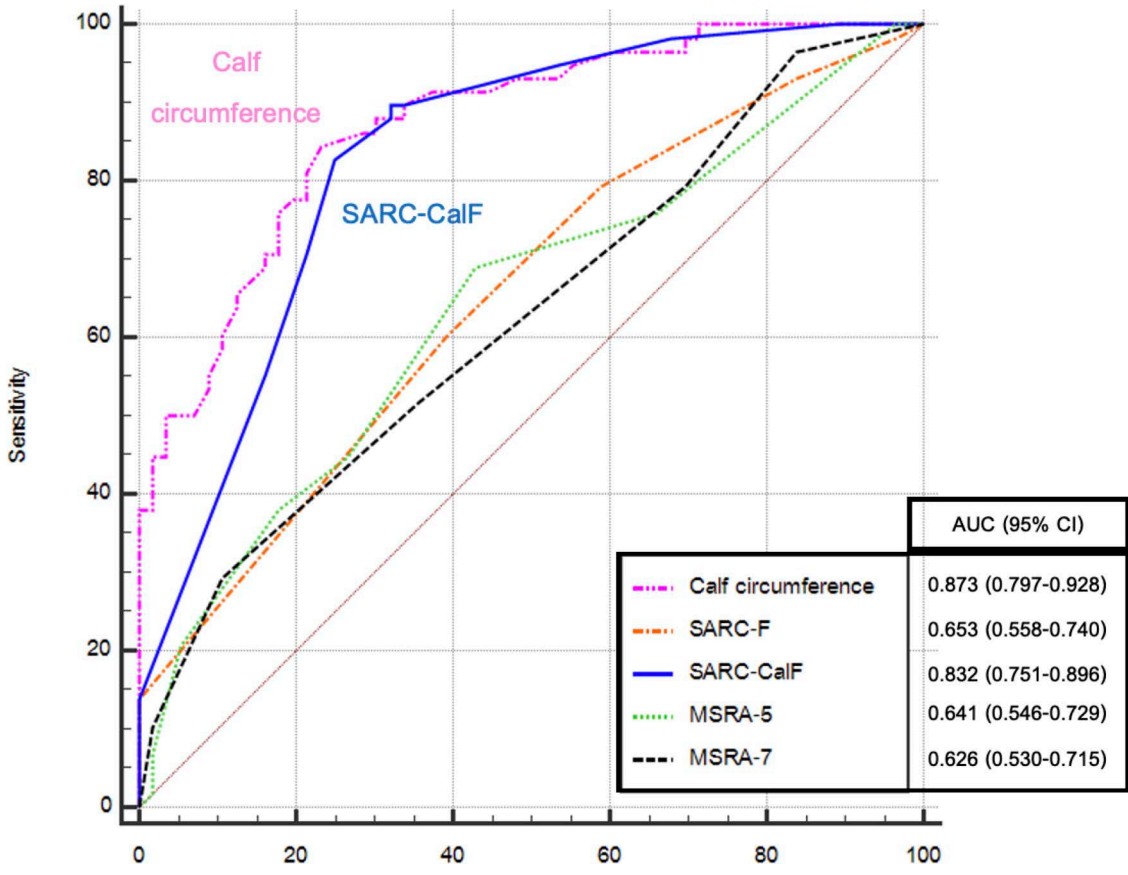

**Fig 2. Receiver operating characteristic curve analysis of the calf circumference; the SARC-F; the SARC-CalF; the MSRA-5 and the MSRA-7 for the diagnosis of sarcopenia among older adults with stroke undergoing inpatient rehabilitation.**

MSRA-5, and MSRA-7 demonstrated poor diagnostic performance and may be unsuitable for sarcopenia screening in this population.

Regarding the CC, its area under the curve of 0.873 reflects good diagnostic ability for inpatient stroke populations. Moreover, this study proposed new cut-off points for the CC: < 33 cm for men and < 32 cm for women. These revised thresholds may improve identification of sarcopenia in this setting. By comparison, the original recommended values were < 34 cm for men, and < 33 cm for women [8,29]. Our findings align with prior studies. Laosuwan et al. reported that cut-off values of < 34 cm for men and < 32.5 cm for women were effective for screening sarcopenia among chronic stroke outpatients [11]. Similarly, Inoue and colleagues in 2021 reported that calf circumference is a reliable screening tool for elderly patients with stroke at inpatient setting [21]. The SARC-CalF also performed well in our study, with an overall AUC of 0.832. This finding is consistent with reports from other settings, including chronic stroke outpatient [11] and community-dwelling older adults [22]. Overall, CC showed slightly higher diagnostic accuracy than SARC-CalF in our study, which may be attributable to the characteristics of stroke survivors.[11,30] CC reflected the structural part of lower limb muscle mass [12,31]. In contrast, the functional items in the SARC-F section of the SARC-CalF may be disproportionately influenced by neurological impairments, such as hemiparesis, impaired motor control, or balance problem [12]. Although SARC-CalF did not demonstrate superior diagnostic accuracy in this group, SARC-CalF remains a useful tool that integrates both structural and functional parts, which may provide additional information on functional status in this population [1]

**Table 2. Diagnostic indices of index tests with original cut-off points.**

| | Cut-off | Sensitivity | Specificity | Accuracy | PPV | NPV | PLR | NLR |
|---|---|---|---|---|---|---|---|---|
| Calf circumference | Men < 34 cm | 90.91% (75.67%−98.08%) | 73.33% (54.11%−87.72%) | 82.54% (70.90%−90.95%) | 78.95% (67.23%−87.27%) | 88.00% (70.93%−95.66%) | 3.41 (1.87-6.23) | 0.12 (0.04-0.37) |
| | Women < 33 cm | 84.00% (63.92%−95.46%) | 65.38% (44.33%−82.79%) | 74.51% (60.37%−85.67%) | 70.00% (57.25%−80.26%) | 80.95% (62.39%−91.59%) | 2.43 (1.39-4.23) | 0.24 (0.10-0.63) |
| SARC-F | ≥ 4 | 100% (93.84%−100%) | 0.00% (0.00%−6.38%) | 50.88% (41.35%−60.36%) | 50.88% (50.88%−50.88%) | N/A | 1.00 (1.00-1.00) | N/A |
| SARC-CalF | ≥11 | 89.66% (78.83%−96.11%) | 67.86% (54.04%−79.71%) | 78.95% (70.31%−86.02%) | 74.29% (66.16%−81.02%) | 86.36% (70.31%−86.02%) | 2.79 (1.89-4.12) | 0.15 (0.07-0.33) |
| MSRA-5 | ≤ 45 | 100% (93.84%−100%) | 0.00% (0.00%−6.38%) | 50.88% (41.35%−60.36%) | 50.88% (50.88%−50.88%) | N/A | 1.00 (1.00-1.00) | N/A |
| MSRA-7 | ≤ 30 | 100% (93.84%−100%) | 0.00% (0.00%−6.38%) | 50.88% (41.35%−60.36%) | 50.88% (50.88%−50.88%) | N/A | 1.00 (1.00-1.00) | N/A |

N/A = not applicable.

SARC-F: The strength, assistance with walking, rise from a chair, climb stairs and falls.

SARC-CalF, the combined test of strength, assistance with walking, rising from a chair, climbing stairs and falls with calf circumference.

MSRA-5: The 5-item Mini Sarcopenia Risk Assessment questionnaire.

MSRA-7: The 7-item Mini Sarcopenia Risk Assessment questionnaire.

PPV: positive predictive value; NPV: negative predictive value; PLR: positive likelihood ratio; NLR: negative likelihood ratio.

**Table 3. Diagnostic indices of calf circumference and the SARC-CalF with new cut-off points using Youden's index.**

| | Cut-off | Sensitivity | Specificity | Accuracy | PPV | NPV | PLR | NLR | Youden's index |
|---|---|---|---|---|---|---|---|---|---|
| Calf circumference: men | < 33 cm | 84.85% (68.10%−94.89%) | 86.67% (69.28%−96.24%) | 85.71% (74.61%−93.25%) | 87.50% (73.54%−94.63%) | 83.87% (69.62%−92.19%) | 6.36 (2.53-16.03) | 0.17 (0.08-0.40) | 0.7152 |
| Calf circumference: women | < 32 cm | 80.00% (59.30%−93.17%) | 76.92% (56.35%−91.03%) | 78.43% (64.68%−88.71%) | 76.92% (61.66%−87.35%) | 80.00% (63.98%−90.01%) | 3.47 (1.67-7.18) | 0.26 (0.12-0.59) | 0.5692 |
| SARC-CalF | ≥ 16 | 82.76% (70.57%−91.41%) | 75.00% (61.63%−85.61%) | 78.95% (70.31%−86.02%) | 77.42% (68.21%−84.56%) | 80.77% (70.08%−88.28%) | 3.31 (2.07-5.29) | 0.23 (0.13-0.41) | 0.5776 |

SARC-CalF: the combined test of strength, assistance with walking, rising from a chair, climbing stairs and falls with calf circumference.

PPV: positive predictive value; NPV: negative predictive value; PLR: positive likelihood ratio; NLR: negative likelihood ratio.

In contrast, the other tools including the SARC-F, MSRA-5, and MSRA-7 demonstrated limited diagnostic value in our population, as evidenced by very low specificity and overall areas under the curves below 0.7. These findings are consistent with previous reports that showed similar limitations for these tools in inpatient and hospital settings [11,24,32,33]. Our study reported that the SARC-F, MSRA-5, and MSRA-7 when applied using with their original cut-off points showed 100% sensitivity but 0% specificity. This is likely because these tools include ambulation- and transfer-related items which are common impairments among stroke inpatients, therefore leading to high false-positive rates.

## Strengths and limitations

Our findings support the use of the CC and the SARC-CalF as practical and accurate tools for sarcopenia screening in inpatient stroke rehabilitation. These tools are particularly valuable in clinical settings with limited access to advanced diagnostic equipment. For example, the CC may also be suitable for other difficult-to-assess populations, including

individuals with cognitive impairment, communication problems, or limited cooperation [34]. In addition, the SARC-CalF incorporates functional assessment while maintaining diagnostic accuracy, making it useful in multiple clinical settings.

Nevertheless, this study had several limitations. First, population specificity: the findings are limited to older inpatients with stroke undergoing rehabilitation and may not be generalizable to other population groups. Second, muscle mass measurement – the bioelectrical impedance analysis was used instead of gold standard techniques such as computed tomography, magnetic resonance imaging, or dual energy X-ray absorptiometry. However, previous studies have shown good agreement between the bioelectrical impedance analysis and the dual energy X-ray absorptiometry in patients with stroke [35]. Third, physical performance was excluded from this study because of patients' mobility limitations. According to the AWGS2019 criteria, sarcopenia can be diagnosed based on the presence of low muscle mass and low muscle strength alone, which is appropriate for this clinical setting [8,18,19]. However, the exclusion of physical performance may lead to differences in prevalence estimates compared with studied strictly applying the AWGS2019 criteria. Our approach may slightly overestimate sarcopenia prevalence by not distinguishing between sarcopenia and severe sarcopenia. Moreover, the diagnostic accuracy of the screening tools should be cautiously interpreted in the context of this adapted operational definition [8]. Importantly, the AWGS2025 guideline defines the main diagnostic components of sarcopenia as low muscle mass and low muscle strength. In addition, physical performance measures are considered outcome indicators rather than required diagnostic components [36]. Fourth, stratified analyses based on stroke phase would demonstrate valuable insights. Nevertheless, the sample size in each stroke phase subgroup in this study was relatively small, which limited the statistical power for meaningful stratified analyses. Future studies with sufficient sample sizes and predefined stratified analytical plan according to stroke phase are needed. Fifth, stroke severity (e.g., National Institute of Health Stroke Scale: NIHSS) was not evaluated, as patients were recruited from a rehabilitation ward, where patients were medically and neurologically stable and beyond the very acute phase of stroke.

## Conclusion

The CC and the SARC-CalF are effective and practical tools for screening sarcopenia among older adults with stroke undergoing inpatient rehabilitation. The optimal cut-off points of CC were < 33 cm in men, and < 32 cm in women. Additionally, the optimal cut-off points of the SARC-CalF were 16 or higher. These tools are especially valuable in settings without access to advanced equipment for measuring muscle mass or strength.

## Acknowledgments

The authors would like to thank Miss Julaporn Pooliam from the Siriraj Research Data Management Unit, Research Development Division, Research Department, Faculty of Medicine Siriraj Hospital, Mahidol University, for her valuable suggestions on data analysis. Moreover, the authors are thankful to the staff for their dedication to this research.

## Author contributions

**Conceptualization:** Kanokphol Supasirimontri, Vilai Kuptniratsaikul.

**Data curation:** Kanokphol Supasirimontri, Metita Wiwitkul.

**Formal analysis:** Kanokphol Supasirimontri, Metita Wiwitkul, Vilai Kuptniratsaikul.

**Funding acquisition:** Vilai Kuptniratsaikul.

**Investigation:** Kanokphol Supasirimontri, Metita Wiwitkul.

**Methodology:** Kanokphol Supasirimontri, Metita Wiwitkul, Vilai Kuptniratsaikul.

**Project administration:** Kanokphol Supasirimontri, Vilai Kuptniratsaikul.

**Resources:** Kanokphol Supasirimontri, Vilai Kuptniratsaikul.

**Supervision:** Vilai Kuptniratsaikul.

**Validation:** Vilai Kuptniratsaikul.

**Writing – original draft:** Kanokphol Supasirimontri, Metita Wiwitkul, Vilai Kuptniratsaikul.

**Writing – review & editing:** Kanokphol Supasirimontri, Metita Wiwitkul, Vilai Kuptniratsaikul.

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
