## [Decision Letter · Decision Letter 0]

28 Dec 2025

PONE-D-25-55631Accuracy of sarcopenia screening tools among older adults with stroke undergoing inpatient rehabilitationPLOS One

Dear Dr. Kuptniratsaikul, Thank you for submitting your manuscript to PLOS ONE. After careful consideration, we feel that it has merit but does not fully meet PLOS ONE’s publication criteria as it currently stands. Therefore, we invite you to submit a revised version of the manuscript that addresses the points raised during the review process.

We look forward to receiving your revised manuscript.

Kind regards,

Francesco Curcio, M.D., Ph.D.

Academic Editor

PLOS One

Journal Requirements:

“Faculty of Medicine Siriraj Hospital, Mahidol University, Bangkok, Thailand (Grant Number R016631042.).”

5. We note that your Data Availability Statement is currently as follows: All relevant data are within the manuscript and its Supporting Information files.

6. Please include a separate caption for each figure in your manuscript.

Additional Editor Comments:

Based on the reviewers’ feedback, the manuscript would benefit from major revisions. We kindly invite the authors to carefully address the comments and suggestions in order to strengthen the quality and clarity of the work.

Reviewers' comments:

Reviewer's Responses to Questions

Comments to the Author

1. Is the manuscript technically sound, and do the data support the conclusions?

Reviewer #1: Yes

Reviewer #2: Yes

2. Has the statistical analysis been performed appropriately and rigorously? 

Reviewer #1: Yes

Reviewer #2: Yes

3. Have the authors made all data underlying the findings in their manuscript fully available?

Reviewer #1: No

Reviewer #2: No

4. Is the manuscript presented in an intelligible fashion and written in standard English?

Reviewer #1: Yes

Reviewer #2: No

5. Review Comments to the Author

Reviewer #1: - In the Abstract background, the statement of an appropriate tool for screening sarcopenia is unknown is not entirely accurate. I think there are appropriate tools but how well it predicts, and eventually diagnose sarcopenia is unknown.

- The Introduction could benefit from expanding the importance of sarcopenia among people with stroke. Since the study later categorises the cohort into acute, subacute and chronic phase, please justify why this differentiation is needed in Intro or Discussion

- In the Methods, please define stable medical conditions as one of your eligibility criteria.

- Was the FAC assessment of their current ability or their pre-stroke ability

- How did the assessment take into account the person’s disability due to their stroke? For example, stroke causing hemiparesis and assessing hand grip strength. What about issues of dysphasia?

- Some details about the stroke would be good to put findings in some context – infarct/haemorrhagic, severity, etc.

- Why do you think CC and SARC-CalF both exhibit similar performance? Should we just do CC?

- Do you think not assessing physical performance impacted findings? Especially in AWGS 2019, this contributed to sarcopenia diagnosis

- Consider how the new AWGS 2025 fits into this

- Sentence structure that could be improved

Reviewer #2: This study evaluates the diagnostic accuracy of five sarcopenia screening tools—calf circumference (CC), SARC-F, SARC-CalF, MSRA-5, and MSRA-7—in an understudied yet clinically important population: older adults with stroke undergoing inpatient rehabilitation. The study addresses a relevant knowledge gap; however, several critical issues need to be considered.

- The study defines sarcopenia based on low muscle mass and low muscle strength, while excluding physical performance. Although this choice is understandable in stroke inpatients, it represents a modification of the AWGS2019 criteria. The authors should explicitly state that an adapted operational definition was used and discuss its potential impact on prevalence estimates and the diagnostic performance of the tools.

- Given the heterogeneity in stroke phase (acute, subacute, and chronic), stratified analyses would be necessary to assess whether diagnostic accuracy varies across these subgroups.

- The manuscript contains some grammatical errors that require careful revision.

6. PLOS authors have the option to publish the peer review history of their article (what does this mean?). If published, this will include your full peer review and any attached files.

Do you want your identity to be public for this peer review? For information about this choice, including consent withdrawal, please see our Privacy Policy.

Reviewer #1: No

Reviewer #2: No

---

## [Author Response · Author response to Decision Letter 1]

13 Feb 2026

9th February 2026

Dear Academic Editor and Reviewers,

We sincerely thank the Academic Editor and the reviewers for their careful review of our manuscript entitled “Accuracy of sarcopenia screening tools among older adults with stroke undergoing inpatient rehabilitation” and for their constructive and insightful comments. We have revised the manuscript accordingly and believe that these revisions have improved its quality and clarity.

Response to Reviewer #1

Comment 1:

In the Abstract background, the statement of an appropriate tool for screening sarcopenia is unknown is not entirely accurate. I think there are appropriate tools but how well it predicts, and eventually diagnose sarcopenia is unknown.

Response:

Thank you for this valuable comment. Our response was presented in the Abstract section (Page 2, Lines 3-5) as follow:

“Appropriate screening tools for sarcopenia are available for older adults with stroke undergoing inpatient rehabilitation, however how well these tools predict and eventually diagnose sarcopenia remain unknown.”

Comment 2:

The Introduction could benefit from expanding the importance of sarcopenia among people with stroke. Since the study later categorises the cohort into acute, subacute and chronic phase, please justify why this differentiation is needed in Intro or Discussion

Response:

Our response was presented in the Introduction section (Page 4, Lines 5-11) as follow:

“Sarcopenia can occur across all phases of stroke.[3] A previous meta-analysis found that half of patients with stroke develop sarcopenia.[4] The prevalence of sarcopenia increases from pre-stroke period through the acute phase and continues to rise during the first month after stroke.[5] Moreover, the prevalence of sarcopenia tends to be higher in the chronic phase of stroke than in the acute phase.[3] Sarcopenia among patients with stroke can adversely affect functional outcomes such as ambulation, activities of daily living, and swallowing function.[6-7]”

Comment 3:

In the Methods, please define stable medical conditions as one of your eligibility criteria.

Response:

Our response was presented in the Materials and methods section (Page 5, Lines 12-15) as follow:

“Stable medical and neurological conditions were defined as the absence of acute events requiring urgent intervention and eligibility for inpatient rehabilitation. In addition, functional and cognitive readiness were evaluated by the ability to maintain an upright position for a minimum of two hours and intact recent memory within the past twenty-four hours [13]”

Comment 4:

Was the FAC assessment of their current ability or their pre-stroke ability?

Response:

Our response was presented in the Materials and methods section (Page 6, Lines 7-9) as follow:

“Demographic data were collected through face-to-face interview with eligible participants and by reviewing their medical records. Stroke-related data included comorbidities, stroke characteristics, and the current Functional Ambulation Categories (FAC) at admission.”

Comment 5:

How did the assessment take into account the person’s disability due to their stroke? For example, stroke causing hemiparesis and assessing hand grip strength. What about issues of dysphasia?

Response:

Our response for handgrip strength was presented in the Materials and methods section (Page 7, Lines 13-17) as follow:

“Handgrip strength of each participant was assessed bilaterally to account for stroke-related motor impairments including unilateral and bilateral upper limb weakness. Three measurements were obtained for each hand, and the mean value of the three trials was calculated for each side. The higher mean handgrip strength was selected as the representative value for subsequent analyses.[8, 12, 20]”

Our response for dysplasia was presented in the Materials and methods section (Page 5, Lines 17-18).

“(1) inability to communicate in the Thai language or the presence of communication impairments, including all types of dysphasia”

Comment 6:

Some details about the stroke would be good to put findings in some context – infarct/haemorrhagic, severity

Response:

Our response for details of stroke type was presented in the Results section (Page 10, Lines 18-19) as follow:

“Eighty-seven patients (76.32%) had ischemic stroke.”

Concerning for the details of severity, our response was presented in the Limitation of Discussion section (Page 18, Lines 9-12).

“Fifth, stroke severity (e.g., National Institute of Health Stroke Scale: NIHSS) was not evaluated, as patients were recruited from a rehabilitation ward, where patients were medically and neurologically stable and beyond the very acute phase of stroke.”

Comment 7:

Why do you think CC and SARC-CalF both exhibit similar performance? Should we just do CC?

Response:

As both tools had different benefits, our study discussed the advantage of each tool. Our response was presented in the Discussion section (Page 16, Lines 11-18) as follow:

“Overall, CC showed slightly higher diagnostic accuracy than SARC-CalF in our study, which may be attributable to the characteristics of stroke survivors.[11, 30] CC reflected the structural part of lower limb muscle mass.[12, 31] In contrast, the functional items in the SARC-F section of the SARC-CalF may be disproportionately influenced by neurological impairments, such as hemiparesis, impaired motor control, or balance problem.[12] Although SARC-CalF did not demonstrate superior diagnostic accuracy in this group, SARC-CalF remains a useful tool that integrates both structural and functional parts, which may provide additional information on functional status in this population.[1]”

Comment 8:

Do you think not assessing physical performance impacted findings? Especially in AWGS 2019, this contributed to sarcopenia diagnosis

Response:

Our response was presented in the Discussion section (Page 17, Lines 21-23, and page 18, Lines 1-3.” As follow:

“However, the exclusion of physical performance may lead to differences in prevalence estimates compared with studied strictly applying the AWGS2019 criteria. Our approach may slightly overestimate sarcopenia prevalence by not distinguishing between sarcopenia and severe sarcopenia. Moreover, the diagnostic accuracy of the screening tools should be cautiously interpreted in the context of this adapted operational definition.[8]”

Comment 9:

Consider how the new AWGS 2025 fits into this

Response:

Concerning the AWGS2025, our response was presented in the Discussion section (Page 18, Lines 3-5) as follow:

“Importantly, the AWGS2025 guideline defines the main diagnostic components of sarcopenia as low muscle mass and low muscle strength. In addition, physical performance measures are considered outcome indicators rather than required diagnostic components.[36]”

Comment 10:

Sentence structure that could be improved

Response:

Thank you for your comment. We rechecked all the sentence structure through the manuscript as suggestion.

Response to Reviewer #2

Comment 1:

The study defines sarcopenia based on low muscle mass and low muscle strength, while excluding physical performance. Although this choice is understandable in stroke inpatients, it represents a modification of the AWGS2019 criteria. The authors should explicitly state that an adapted operational definition was used and discuss its potential impact on prevalence estimates and the diagnostic performance of the tools.

Response:

Our response was presented in the Discussion section (Page 17, Lines 21-23, and page 18, Lines 1-3.” As follow:

“However, the exclusion of physical performance may lead to differences in prevalence estimates compared with studied strictly applying the AWGS2019 criteria. Our approach may slightly overestimate sarcopenia prevalence by not distinguishing between sarcopenia and severe sarcopenia. Moreover, the diagnostic accuracy of the screening tools should be cautiously interpreted in the context of this adapted operational definition.[8]”

Comment 2:

Given the heterogeneity in stroke phase (acute, subacute, and chronic), stratified analyses would be necessary to assess whether diagnostic accuracy varies across these subgroups.

Response:

Thank you for your comment. Our response was presented in the Discussion section (Page 18, Lines 6-9) as follow:

“Fourth, stratified analyses based on stroke phase would demonstrate valuable insights. Nevertheless, the sample size in each stroke phase subgroup in this study was relatively small, which limited the statistical power for meaningful stratified analyses. Future studies with sufficient sample sizes and predefined stratified analytical plan according to stroke phase are needed.”

Comment 3:

The manuscript contains some grammatical errors that require careful revision.

Response:

Thank you for your comment. We obtained the grammatical checking through the manuscript as suggestion.

Response to Academic Editor

Comment 1:

Response:

We thank the Academic Editor for this suggestion and have revised the manuscript accordingly. We confirm that the manuscript has been prepared in full compliance with the PLOS ONE’s style requirements, including file naming, and follows the official journal template provided on the website.

Comment 2:

We note that the grant information you provided in the ‘Funding Information’ and ‘Financial Disclosure’ sections do not match.

Response:

Thank you for the notification. We have carefully reviewed the funding information and corrected the grant numbers so that the information in the Funding Information and Financial Disclosure sections is now consistent and accurate.

Comment 3:

Response:

We thank the Academic Editor for this suggestion and have revised the manuscript accordingly. We confirm that the funders had no role in study design, data collection and analysis, decision to publish, or preparation of the manuscript. This statement has been included in the cover letter as requested.

Comment 4:

When completing the data availability statement of the submission form, you indicated that you will make your data available on acceptance. We strongly recommend all authors decide on a data sharing plan before acceptance, as the process can be lengthy and hold up publication timelines. Please note that, though access restrictions are acceptable now, your entire data will need to be made freely accessible if your manuscript is accepted for publication. This policy applies to all data except where public deposition would breach compliance with the protocol approved by your research ethics board. If you are unable to adhere to our open data policy, please kindly revise your statement to explain your reasoning and we will seek the editor's input on an exemption. Please be assured that, once you have provided your new statement, the assessment of your exemption will not hold up the peer review process.

Response:

Thank you for the clarification. We confirm that, upon acceptance of the manuscript, the dataset underlying the findings will be made publicly available in accordance with the journal’s open data policy. We have updated the Data Availability Statement accordingly.

Comment 5:

We note that your Data Availability Statement is currently as follows: All relevant data are within the manuscript and its Supporting Information files.

Please confirm at this time whether or not your submission contains all raw data required to replicate the results of your study. Authors must share the “minimal data set” for their submission.

Response:

We thank the Academic Editor for this suggestion and have revised the manuscript accordingly.

We confirm that the submission contains raw data required to replicate the results of the study. The minimal data set necessary for replication is fully included within the manuscript and its Supporting Information files.

Comment 6:

Please include a separate caption for each figure in your manuscript.

Response:

We have revised the manuscript to include a separate caption for each figure, in accordance with the journal’s requirements.

Comment 7:

Response:

We thank the Academic Editor for this suggestion and have revised the manuscript accordingly.

We have carefully reviewed the publications as suggested by the reviewer. Those deemed relevant to the scope and findings of our study have been cited in the revised manuscript.

We hope that our responses adequately address all comments and that the revised manuscript meets the expectations of the Academic Editor and reviewers. We sincerely appreciate the opportunity to revise our work.

Sincerely,

Assoc. Prof. Vilai Kuptniratsaikul

---

## [Decision Letter · Decision Letter 1]

24 Mar 2026

Accuracy of sarcopenia screening tools among older adults with stroke undergoing inpatient rehabilitation

PONE-D-25-55631R1

Dear Dr. Vilai Kuptniratsaikul ,

We’re pleased to inform you that your manuscript has been judged scientifically suitable for publication and will be formally accepted for publication once it meets all outstanding technical requirements.

Kind regards,

Francesco Curcio, M.D., Ph.D.

Academic Editor

PLOS One

Additional Editor Comments (optional):

We have no additional comments and are in agreement with the reviewers. We believe that the manuscript has improved significantly as a result of the revisions.

Reviewers' comments:

Reviewer's Responses to Questions

Comments to the Author

1. If the authors have adequately addressed your comments raised in a previous round of review and you feel that this manuscript is now acceptable for publication, you may indicate that here to bypass the “Comments to the Author” section, enter your conflict of interest statement in the “Confidential to Editor” section, and submit your "Accept" recommendation.

Reviewer #1: All comments have been addressed

Reviewer #2: All comments have been addressed

2. Is the manuscript technically sound, and do the data support the conclusions?

Reviewer #1: Yes

Reviewer #2: (No Response)

3. Has the statistical analysis been performed appropriately and rigorously? 

Reviewer #1: Yes

Reviewer #2: (No Response)

4. Have the authors made all data underlying the findings in their manuscript fully available?

Reviewer #1: Yes

Reviewer #2: (No Response)

5. Is the manuscript presented in an intelligible fashion and written in standard English?

Reviewer #1: Yes

Reviewer #2: (No Response)

6. Review Comments to the Author

Reviewer #1: No further comments. All comments appear to have been addressed by the authorship team. I have no further input

Reviewer #2: (No Response)

7. PLOS authors have the option to publish the peer review history of their article (what does this mean?). If published, this will include your full peer review and any attached files.

Do you want your identity to be public for this peer review? For information about this choice, including consent withdrawal, please see our Privacy Policy.

Reviewer #1: No

Reviewer #2: No

---

## [Editor Report · Acceptance letter]

PONE-D-25-55631R1

PLOS One

Dear Dr. Kuptniratsaikul,

I'm pleased to inform you that your manuscript has been deemed suitable for publication in PLOS One. Congratulations! Your manuscript is now being handed over to our production team.

Kind regards,

on behalf of

Dr. Francesco Curcio

Academic Editor

PLOS One